

# Persistent growth of anthropogenic NMVOC emissions in China during 1990-2017: dynamics, speciation, and ozone formation potentials

Meng Li[1,2], Qiang Zhang[1], Bo Zheng[3], Dan Tong[1], Yu Lei[4], Fei Liu[3], Chaopeng Hong[1], Sicong Kang[3], Liu Yan[1], Yuxuan Zhang[1], Yu Bo[6], Hang Su[5,2], Yafang Cheng[5,2] and Kebin He[3,1]

[1] Ministry of Education Key Laboratory for Earth System Modeling, Department of Earth System Science, Tsinghua University, Beijing 100084, China

[2] Max Planck Institute for Chemistry, Mainz 55128, Germany

[3] State Key Joint Laboratory of Environment Simulation and Pollution Control, School of Environment, Tsinghua University, Beijing 100084, China

[4] China Academy for Environmental Planning, Beijing 100012, China

[5] Center for Air Pollution and Climate Change Research, Jinan University, Guangzhou 511443, China

[6] Key Laboratory of Regional Climate-Environment for Temperate East Asia, Institute of Atmospheric Physics, Chinese Academy of Science, Beijing 100029, China

*Correspondence to*: Qiang Zhang (qiangzhang@tsinghua.edu.cn)

**Abstract.** Non-methane volatile organic compounds (NMVOC) are important ozone and secondary organic aerosol precursors and play important roles in tropospheric chemistry. In this work, we estimated the total and speciated NMVOC emissions from China's anthropogenic sources during 1990-2017 by using a bottom-up emission inventory framework, and investigated the main drivers behind the trends. We found that, anthropogenic NMVOC emissions in China have been increased continuously since 1990 due to the dramatic growth in activity rates and absence of effective control measures. We estimated that, anthropogenic NMVOC emissions in China increased from 9.76 Tg in 1990 to 28.5 Tg in 2017, mainly driven by persistent growth from the industry sector and solvent use. In the meanwhile, emissions from the residential and transportation sectors declined after 2005, partly offset the total emission increase. During 1990-2017, mass-based emissions of alkanes, alkenes, alkynes, aromatics, oxygenated VOCs (OVOC) and other species increased by 274%, 88%, 4%, 387%, 91%, and 231% respectively. Following the growth in total NMVOC emissions, the corresponding ozone formation potential (OFP) increased form 38.2 Tg-$O_3$ in 1990 to 99.7 Tg-$O_3$ in 2017. We estimated that aromatics accounted for the largest share (43%) of total OFP, followed by alkenes (37%) and OVOC (10%). Growth in China's NMVOC emissions were mainly driven by the transportation sector before 2000, while industrial sector and solvent use dominated the emission growth during 2000-2010. After 2010, although emissions from the industry sector and solvent use kept growing, strict control measures on transportation and fuel transition in residential stoves have successfully slowed down the increase trend,



especially after the implementation of China's clean air action since 2013. However, compared to large emission decreases of other criteria air pollutants in China (e.g., $SO_2$, $NO_x$, and primary PM) during 2013-2017, the relatively flat trend in NMVOC emissions and OFP revealed the absence of effective control measures, which might have contributed to the increase of ozone during the same period. Given their high contributions to emissions and OFP, tailored control measures for solvent use and industrial sources should be developed, and collaborative control strategies should be designed to mitigate both $PM_{2.5}$ and ozone pollution simultaneously.

## 1 Introduction

With rapid economic growth and urbanization, high concentrations of ground ozone and aerosols have caused severe adverse effects on air quality, ecosystems and human health (Monks et al., 2015; Lu et al., 2018). Non-methane volatile organic compounds (NMVOC) play crucial roles in producing ozone and secondary organic aerosols (SOA), and some NMVOC are toxic. NMVOC can be emitted from a variety of sources, including biogenic, open biomass burning, anthropogenic stationary and mobile sources (van der Werf et al., 2010; Guenther et al., 2012; Li et al., 2017). In China, a series of studies have revealed that reducing NMVOC emissions from anthropogenic sources is key to controlling ozone and fine particulate matter ($PM_{2.5}$, with aerodynamic diameters less than or equal to 2.5 μm) pollution in Chinese cities (Shao et al., 2009; Yuan et al., 2013; Jin et al., 2015).

The emission inventory of NMVOC for China has been under development to support atmospheric composition analyses and policy-making (Klimont et al., 2002; Bo et al., 2008; Zhang et al., 2009; Li et al., 2014; Wei et al., 2014; Wu et al., 2016). Despite considering local statistics and measurements, the uncertainties in NMVOC emissions are still high, i.e., ±68% ~ ±78%, due to the scarcity of accurate information for scattered areal sources (e.g., solvent use, unorganized industrial volatilization) (Zhang et al., 2009; Kurokawa et al., 2013). In recent decades, dramatic changes in NMVOC emissions have taken place in China driven by economic development as well as the implementation of control measures for key sectors of industry, residential and transportation (Wu et al., 2016; Li et al., 2017; Zheng et al., 2018). Energy-efficient and environmentally friendly technologies have entered the market, and the shares are estimated to be increasing. In addition, China has implemented stringent clean air policies to mitigate pollutants in recent years, driving the significant emission reductions of related criteria pollutants during 2013-2017 (Zheng et al., 2018).

Considering the large variances in chemical reactivity for different species, long-term chemically-resolved emissions are urgently needed for targeted air quality management in China. Tremendous efforts have been made to estimate the speciated NMVOC emissions to feed chemical transport models (CTMs) and composition analyses (Zhang et al., 2009; Li et al., 2014; Wu and Xie, 2017). Li et al. (2014) developed emissions with speciation in 2006 based on the compiled source profile database. This work pointed out that profiles are the major uncertainty sources for model-ready emissions in China. A complete source profile database for China is still under development. Ground flux measurements confirmed the importance of oxygenated volatile organic compound (OVOC) to reconcile between model simulations and observations (Karl et al.,



2018). Mo et al. (2016) compiled an updated source profile database for hydrocarbons and OVOC by including local profiles measured in recent years. However, few studies have used this information on NMVOC speciation for long-term chemical analyses. Recent studies quantifying emissions of NMVOC have only focused on a specific year (e.g., Li et al., 2014; Wu and Xie, 2017) or have lacked speciation (e.g., Wu et al., 2016) and thus still lag behind the needs for regulatory decision

making. To support the chemical analyses from the past to present, we developed long-term speciated NMVOC emissions for all anthropogenic sources in China in this work, covering alkanes, alkenes, aromatics, alkynes, OVOC, and halocarbons for the period of 1990 to 2017.

The increasingly severe ozone pollution in China has been observed by the national monitoring network since 2013 (Li K et al., 2018; Lu et al., 2018). Lu et al. (2018) showed that the exposure of humans and vegetation to ozone in China is highest

among the developed regions of the world. Li K et al. (2018) resolved the effects of meteorological variability, emissions and multiphase chemistry on the ozone trends based on model simulations. Identifying the reasons of surface ozone rise is crucial to design ozone control policy and protect human health and ecosystems. In context of significant reductions for criteria pollutants such as $NO_2$, $SO_2$, CO and $PM_{2.5}$ attributed to the implementation of control measures (Zheng et al., 2018), the trend of NMVOC emissions and their potential effects on ozone production is key to understand the observed ozone

trend and establish mitigation measures in the near future. To address this issue, we calculated OFP (ozone formation potential) as an indicator based on the speciated NMVOC emissions.

## 2    Methods and data

### 2.1    NMVOC emissions

We estimated the emissions of NMVOC from 1990 to 2017 following the bottom-up framework of the Multi-resolution

Emission Inventory for China (MEIC) model (available at www.meicmodel.org). The emissions were calculated based on a technology-based methodology, as described in detail by earlier papers (Zhang et al., 2009; Zheng et al., 2014; Liu et al., 2015; Li et al., 2017; Zheng et al., 2018). Briefly, emissions for stationary sources were estimated based on the "emission factor" method following equation (1):

$$E_i = \sum_j \sum_k A_{i,j,k} [\sum_m X_{i,j,k,m} EF] \tag{1}$$

where $i$ denotes the administrative unit of China, $j$ is the source category in the classification system, $k$ represents the fuel type for combustion-related sources or product for industrial processes, $m$ is the technology of fuel combustion or industrial production. $E$ represents the estimated emissions, which is integrated by $A$ (activity rate), $X$ (technology distribution ratio) and $EF$ (emission factor) for each emitting source. $EF$ is determined based on the raw unabated emission factor ($EF_{raw}$), the penetration ratio ($C_n$) and the removal efficiency of the control technology $n$ as follows:

$$EF = EF_{raw} \sum_n C_n (1-\eta_n) \tag{2}$$




For power plants, NMVOC emissions were derived from the China coal-fired Power Plant Emissions Database (CPED, Liu et al., 2015), which is developed based on detailed information on fuel type, fuel quality, combustion technology, and pollutant abatement facilities for >7600 power generation units nationwide in China. The improved emissions for the on-road transportation sector developed by Zheng et al. (2014) were integrated into the framework of MEIC, which estimated the vehicle population and emission factors at a county level.

A detailed four-level source classification system, representing sector, fuel/product, technology/solvent type and end-of-pipe pollutant abatement facilities, was established by including over 700 emitting sources in the MEIC model. Only anthropogenic sources, excluding open biomass burning, aviation and international shipping, were considered. We present a total of 5 sectors (power, industry, residential, solvent use and transportation) and 15 subsectors by combining 109 NMVOC emitting sources by fuel type, industrial product, solvent use, vehicle type and diesel engine in Table 1. The detailed source categories, activity rates, emission factors and references are given in Table S1. Both combustion processes using fossil fuel and biofuel were considered for boilers and stoves. The subsector of "oil production, distribution and refinery" includes the evaporative emissions during oil production, transfer, refining, and refueling in oil stations.

Paint use was further divided from the solvent use sector and includes the paint use applied for architecture, vehicles, wood, and other industrial purposes. The inter-annual market shares of waterborne and solvent-based paint were further taken into account for each source category. Printing ink, pharmaceutical production, pesticide use, glue use, and domestic solvent use were separately calculated and grouped into the subsector of "solvent use other than paint".

For on-road transportation, we set up a process-based calculation framework for gasoline and diesel vehicles classified into eight types covering both trucks and passenger cars and four performance categories (high-duty, medium-duty, light-duty and mini). Each emitting process, including pollutant exhaust in running mode and NMVOC evaporation, was considered. China's emission standards covering pre-Euro I and Euro I to Euro V during 1990-2017 were applied for each vehicle type as listed in Zheng et al. (2018).

### 2.1.1 Activity rates

Activity rates during 1990-2017 were mainly gathered and assigned from various official statistics released by the National Bureau of Statistics (NBS). The inter-annual coal consumption rates for each power plant unit were obtained from the CPED database (Liu et al., 2015). For combustion-related sources in the industrial and residential sectors, the consumption rates of fossil fuel and biofuel were obtained from the provincial energy balance spreadsheets of the China Energy Statistical Yearbook (NBS, 1992-2017). The yields of industrial products were collected from various officially released statistics for the specific economic sector, such as China Statistical Yearbook, China Rubber Industry Yearbook, China Chemical Industry Yearbook, China Light Industry Yearbook, China Paint and Coatings Industry Yearbook; China National Petroleum Corporation Yearbook, China Trade and External Economic Statistical Yearbook, China Plastics Industry Yearbook, China Industry Economy Statistical Yearbook, China Sugar & Liquor Yearbook, and China Food Industry Yearbook (for references, see Table S1 ).



The amounts of solvent use were gathered or estimated from a wide range of available statistics and peer-reviewed literature published by Chinese researchers (China Paint and Coatings Industry Yearbook; China Chemical Industry Yearbook; China Industry Economy Statistical Yearbook; China Forestry Statistical Yearbook; China Statistical Yearbook for Regional Economy; Wei et al., 2009). Paint use was further divided into 7 subcategories (as listed in Table S1) by assigning a splitting
ratio based on local studies (Wei et al., 2009). For solvent use other than paint, the solvent consumption amounts were obtained from statistics or reports (for printing, vehicle treatment, wood production, pharmaceutical production, pesticide use, dry clean, glue use), or estimated using proxies (for domestic solvent). Limited information by provinces are available for the solvent use sector, we allocated the national activity rates derived from yearbooks into provinces based on the construction area, vehicle production, vehicle ownership, cultivation area, and etc., according to the solvent application type.

The activity rates of on-road vehicles were assigned following the approach of Zheng et al. (2014), which modeled the vehicle ownership and fuel consumptions by counties for each vehicle type, with provincial statistics as inputs (NBS, 2000-2015; NBS, 1990-2017). The diesel amounts consumed in off-road engines for each province were obtained from the sector-specific statistical data (China Transportation & Communications Yearbook; China Automotive Industry Yearbook; China Agriculture Statistical Report; China Statistical Yearbook on Construction).

**2.1.2 Emission factors**

Emission factors were determined based on first-hand measurements or local surveys, including the European Environment Agency (EEA) guidebook (EEA, 2016), the AP-42 database (EPA, 1995), and peer-reviewed literature (Tsai et al., 2003; He, 2006; Wei et al., 2009; Zheng et al., 2014). The unabated emission factors used in our calculation and their references are listed in Table S1. Previous studies have conducted a comprehensive overview of available emission factors from
measurements in China and databases from other countries that complied with China's inventory compiling system (Klimont et al., 2002; Bo et al., 2008; Wu et al., 2016). We firstly evaluated the emission factors based on local measurements or determined by taking China's regulations into account, e.g., the values of Wei et al. (2009) for solvent use, Tsai et al. (2003) for residential coal combustion, and the technology-based emission factors derived from Zheng et al. (2014) for on-road vehicles. For sources that lack reliable local emission factors, we mainly refer to European studies (EEA, 2016) or AP-42
(EPA, 1995), combined with source information from local investigations where available (Zhang et al., 2000; Tsai et al., 2003; He, 2006; Li et al., 2011; Wang et al., 2013).

Control strategies for NMVOC have been applied to solvent use, industry, residential and transportation sources in recent years. The underlying technology penetration rates were derived from reports and surveys and supplemented with unpublished data from the Ministry of Ecology and Environment of the People's Republic of China (Zheng et al., 2018;
Peng et al., 2019). As of 2017, a series of regulations on paint use covering wood, architecture, industrial, and vehicle applications have been established nationwide, leading to the decline of corresponding emission factors. To comply with the emission standard in GB 18582-2008, waterborne paint containing low levels of organic chemicals has dominated the architectural interior wall coating since 2008 (Wei et al., 2009). Proportions of waterborne paints applied in architectural



outdoor and automobile production lines have increased gradually, with changes of 15% to 84%, 5% to 37% during 2005-2017 respectively according to local surveys (Wang and Li, 2012). Notably, in Guangdong province, the waterborne solvent products have dominated the local market covering a wide range of industries (with shares ≥ 70%) by 2017, benefited from the pioneering implementation of environmental legislations. Replacing coal with natural gas and electricity in both industrial and residential boilers, and fuel transitions from biofuel to commercial energy driven by increasing per capita income have decreased the average emission strength (Peng et al., 2019). The stage-by-stage stringent emission standards implemented for on-road vehicles have had substantial effects on NMVOC emission reduction (Zheng et al., 2018). Newly registered vehicles must comply with the latest emission standards. Following the timeline of standards release, gasoline light duty vehicles meeting Euro IV and Euro V standards occupied > 55% and >8% respectively nationwide by 2017. In the mean while, the proportions of Euro V have increased up to > 60% in Beijing and Shanghai. For diesel vehicles, the shares of Euro IV were estimated in the range of 29%~63% nationwide in 2017, varying by vehicle duties. In Beijing and Shanghai, diesel vehicles meeting Euro V were estimated to account for 29%~74% in the fleet. In addition, by 2017, all "yellow label" vehicles were eliminated for both gasoline and diesel vehicles, further eliminated NMVOC emissions from super emitters (Zheng et al., 2018).

Regarding OVOC, we corrected the emission factors for on-road vehicles. Because current emission factors are only for non-methane hydrocarbons (NMHCs), we applied correction ratios of 1.32, 1.08, 1.10, and 1.06 for heavy-duty, light-duty diesel vehicles, heavy-duty, and light-duty gasoline vehicles to the original values to comply with the follow-up speciation for the total NMVOC, assuming OVOC fractions of 32%, 8%, 10%, and 6% respectively, following the method of Li et al. (2014) and source profiles listed in Table S1.

## 2.2 Speciation of NMVOC

Following Li et al. (2014), we developed emissions by individual chemical species based on the profile-assignment approach. First, we established a "composite" profile database for China by integrating the local profiles and supplementing it with the SPECIATE v4.5 database for absent sources (Simon et al., 2010, available at: https://www.epa.gov/air-emissions-modeling/speciate-version-45-through-40 ). Then, regarding OVOC, we reviewed the profiles for all combustion-related sources, including the combustion of coal, fuel oil, gasoline, diesel in the power, industry, residential and transportation sectors, and corrected the incomplete profiles that were absent from the OVOC fractions. Especially, OVOC accounts for more than 30% for the residential coal (31%) and biofuel use (23% ~ 33%), and exhaust from heavy duty diesel vehicles (32%). Finally, we assigned the composite profile to each source by setting up the source linkage between the profiles and the inventory. The selected source profiles used in this work are presented in Table S1.

The detailed procedure for developing the composite profile database is illustrated in Li et al. (2014). Briefly, for sources for which local profiles are available and there are significant differences in technology or legislation between China and western countries, only local profiles are used; otherwise, all corresponding profiles are listed as "candidate" ones and included for further compilation in the composite profile database. The gathered local profiles cover major contributing





sources: biofuel combustion (Tsai et al., 2003; Liu et al., 2008; Wang et al., 2009; Mo et al., 2016), coal combustion (Liu et al., 2008; Shi et al., 2015), asphalt production (Liu et al., 2008), oil production, handling and refinery (Liu et al., 2008), vehicle varnish paint (Yuan et al., 2010), printing ink (Yuan et al., 2010; Zheng et al., 2013; Wang et al., 2014), gasoline evaporation (Liu et al., 2008; Zhang et al., 2013; Wu et al., 2015), Poly'Prop production (Mo et al., 2015), gasoline vehicles (Duffy et al., 1999; Schauer et al., 2002; Liu et al., 2008) and diesel vehicles (Schauer et al., 1999; Liu et al., 2008; Yao et al., 2015; Mo et al., 2016). In addition to the source profiles used for speciation in Li et al. (2014), we updated profiles that were newly added in SPECIATE v4.5 and local profiles measured in recent years (Zhang et al., 2013; Wang et al., 2014; Shi et al., 2015; Wu et al., 2015; Yao et al., 2015; Mo et al., 2015, 2016). Profiles for approximately 60 sources were updated, including power plants, paint production, industrial coal use, gasoline evaporation, coke production, biofuel combustion in residential stoves, glue use, paint use, vehicles and off-road diesel engines. Compared to Li et al. (2014), the update in source profiles results in higher mass fractions for alkenes, alkynes, OVOC and lower contributions for aromatics.

Due to the improper sampling and analysis method used in profile measurements, several local profiles lack significant OVOC fractions (Li et al., 2014). We extended the revision to all combustion processes and corrected their profiles by appending the component of "OVOC" with fractions derived from the "complete" profiles for the same source. The equation of the OVOC revision is as follows:

$$X_{revised}(i,j) = \frac{X_{ori}(i,j)}{\sum_j X_{ori}(i,j)} \times (1 - \overline{X_{ovoc}(i,j)})$$

(3)

where $X_{revised}$ is the mass fraction of species $j$ in the revised profile for source $i$, $X_{ori}$ is the original mass fraction; $\overline{X_{ovoc}}$ represents the calculated mean of the OVOC proportion for all candidate profiles that have OVOC measured.

## 2.3 Calculation of OFP

Ozone formation potential (OFP) is a widely used scale to investigate the potential ozone production due to emissions of NMVOC and has been applied to guide the establishment of most cost-effective ozone mitigation measures (e.g., Song et al., 2007; Zheng et al., 2009). OFP for individual chemical species is calculated based on the mass and maximum incremental reactivity (MIR), which scales the ozone production potential for corresponding species:

$$OFP_{i,j,k} = EVOC_{i,k} \times X_{i,j} \times MIR_j$$

(4)

where $i$, $j$, and $k$ represents the source, chemical species and province, respectively. $OFP$ is the ozone formation potential; $EVOC$ is the total NMVOC emission estimate; $X$ represents the mass fraction for species $j$ emitted from source $i$, derived from the composite profiles in this study; and $MIR$ is the maximum incremental reactivity scale for species $j$ (Carter, 1994, 2010).



## 3 Results

### 3.1 Emission trends and driving forces

China's NMVOC emissions have shown a continuously increasing trend. NMVOC emissions were estimated to be 9.8 Tg in
1990, then increased to 14.5 Tg in 2000, 20.3 Tg in 2005, 25.1 Tg in 2010 and 28.5 Tg in 2017, with annual growth rates of
4.0% (1990-2000), 7.0% (2000-2005), 4.3% (2005-2010) and 1.8% (2010-2017). Emissions by sectors and subcategories for
each sector during 1990-2017 are shown in Fig. 1 and Fig. 2, respectively (details in Table 1). Industry (+6.0 Tg, +368%)
and solvent use (+10.7 Tg, +850%) are the main sectors driving the total emission increase during 1990-2017. The
transportation emissions first increased rapidly and then decreased, peaking at 6.5 Tg in 2008. Residential emissions
gradually decreased during the last decade, dominated by biofuel combustion, leading to a 24% emission decline in 2017
than in 1990. As a result, the proportions by sector for national emissions have changed, with growing contributions from
industry (17%-27% for 1990-2017) and solvent use (13%-42%), shrinking contributions from residential (55%-14%), and
stable contributions from the transportation sector (15%-17%).

The emission trends by subcategory for each major sector (industry, residential, solvent use and transportation) are further
illustrated in Fig. 2. Coal combustion, chemical industry and oil-related processes are the main contributors to the industrial
emission changes. Despite the gradual decline in industrial coal use since 2012, industrial processes still show a continuously
increasing trend, driven by the chemical industry. The rapid emission growth of solvent use can be attributed to several
sources, including paint use and other various solvent applications. Both the residential and transportation sectors have
started to decrease in recent years. The significant reductions in crop residue use in residential stoves are consistent with the
socioeconomic development in rural areas of China. Vehicular emissions show a sharp increase from 1990 to 2008, driven
by the increase in vehicle ownership, then decrease gradually because of the stage-by-stage implementation of VOC
abatement measures, especially for gasoline-fueled passenger cars.

### 3.2 Speciated NMVOC emissions

Emissions by individual chemical species were developed based on the total NMVOC emissions (as illustrated above) and
the mass fractions derived from source profiles. Figure 3 (d) presents the sectorial emissions of the top 30 species in
OFP-descending order in 2017, which together account for >80% of the total OFP. Toluene is the largest contributor to OFP
with emission estimated as 3.4 Tg (12.0% of the total), followed by m- and p-xylene (1.4 Tg, 5.1%), ethylene (1.2 Tg, 4.3%),
o-xylene (0.71 Tg, 2.5%), propylene (0.41 Tg, 1.4%) and formaldehyde (0.41 Tg, 1.4%) (in OFP-descending order). The
different distribution patterns between mass and OFP are attributed to the variations among chemical species in the reactivity
scales of MIR. In 2017, toluene, xylenes (including all isomers of xylene, i.e., m- and p-xylene and o-xylene), and
ethylbenzene were dominated by solvent use, while ethylene, propylene and formaldehyde were mainly contributed by
industrial and residential sources.





We present the emissions by sector for these 30 species in 1990, 2000 and 2010, with calculated OFP as references in Fig. 3 (a-c). Notably, aromatics, including toluene and xylenes, showed dramatic emission increases, whereas alkenes (ethylene, propylene) and OVOC (formaldehyde, acetaldehyde) showed moderate changes. For the 1st 10-year period (1990~2000), ethylene contributed approximately 10% by mass and 20~25% by OFP, ranking first among all identified species. Transportation drove up emissions of all chemical species during 1990-2000. In the 2nd 10-year stage (2000-2010), toluene surpassed ethylene, becoming the largest contributor to the total emissions, with an increase in proportion from 6% to 9%. Similar increasing trends were estimated for other aromatic species linked to the solvent use and industry sectors. Solvent use continuously drove up related species such as toluene and xylenes since 2010 due to the increasing demand and relatively limited penetration of control measures. Meanwhile, ethylene, acetylene, and formaldehyde started to decrease as a result of reductions in residential biofuel use and vehicle exhaust.

Figure 4 further illustrates the emission and OFP trends for six representative species (ethane, ethylene, toluene, xylenes, formaldehyde and acetylene) of chemical groups (alkanes, alkenes, aromatics, OVOC and alkynes). Apart from ethane and acetylene, all species are identified with high contributions to ozone formation during 1990-2017 (see Fig. 4b). Sharp growth was estimated for toluene and xylenes, with 6-fold (>2 Tg) higher emissions in 2017 than in 1990, mainly driven by solvent use as illustrated above. The emissions of ethane first increased then showed slight decrease, and experienced a 39% (+178 Gg) increase in 2017 compared to 1990. Ethylene emissions rose rapidly at first 16 years and then declined, increasing by 11% (+117 Gg) from 1990 to 2017. The declining trend of ethylene in recent years can be attributed to the residential combustion activities. In contrast to the overall growing trend, formaldehyde decreased by 25% (-137 Gg), leading to even lower emissions in 2017 than in 1990, because of the reduced use of biofuel in residential stoves.

As shown in Fig. 4, emission fractions by chemical group changed significantly, with reduced proportions of alkenes and OVOC and increased shares of aromatics and alkanes. In 2017, aromatics was the largest contributing chemical group to emissions, accounting for 33% of the total. The mass fractions for alkenes and OVOC gradually decreased from 20% and 23% in 1990 to 13% and 15% in 2017 respectively. Figure 5 decomposes the driving forces of emission changes by chemical group and sector from 1990 to 2017. During 1990-2000, the 48% (+4.6 Tg) emission increase were mainly attributed to alkanes (+1.8 Tg), aromatics (+1.3 Tg) and alkenes (+0.91 Tg) contributed by the transportation sector. Since 2000, activity rates from solvent use and industry grew rapidly along with the economic development, leading to large emission increases of alkanes and aromatics. For the period of 2000-2010, aromatics and alkanes accounted for 36% and 26% of the total emission growth, respectively, dominated by solvent use and industrial sources. Solvent use and industrial processes continuously promoted the emissions of aromatics (+2.8 Tg) and alkanes (+1.5 Tg) in recent years (2010-2017), but the increasing trend was lowered by the declined emissions of transportation and residential sectors (-2.9 Tg) benefited from the penetration of control measures and transition of fuel types. As a result, a much lower emission growth ratio of 13% was estimated for 2010-2017, compared to > 40% increase for previous decades (see Fig. 5).



### 3.3  OFP

The national OFP shows a persistent increasing trend from 38 Tg $O_3$ to 100 Tg $O_3$, at a growth factor of 2.6 during 1990-2017 (Fig. 1). Due to their large emission amounts and high ozone-producing chemical reactivity (scaled by MIR), ethylene, toluene, xylenes and propylene are estimated to be the key NMVOC precursors in ozone formation during the last
decades (see Fig. 3). The rankings of OFP contribution by individual species have changed over time, with increasingly important roles played by reactive aromatic species (toluene, xylenes, ethylbenzene) and decreasing contributions from alkenes (ethylene, propylene, butenes) and OVOC (formaldehyde, acetaldehyde). Specifically, during 1990-2017, toluene, xylenes, 2-methyl-2-butene, ethylbenzene, 2-butene, 1,2,4-trimethylbenzene, propylene, 2-pentene, and formaldehyde contributed most to the OFP trend. Transportation, solvent use and industry sectors were the main contributors to the OFP
increase and partly suppressed by the decline in biofuel use.

As presented in Fig. 4, xylenes (sum of m-xylene, p-xylene and o-xylene) and toluene surpassed ethylene to become the two largest OFP contributors at present. In 2000, alkenes contributed 47% to the total OFP, while aromatics accounted for 24%. Driven by the increasing emissions of aromatics since 2000, the OFP contributions of aromatics (43%) are now even higher than those of alkenes (37%). Alkenes and aromatics together represented 80% of the total OFP in 2017. Among the top 30
species contributors, the OFP proportions of aromatics are even higher, increasing from 20% in 1990 to 50% in 2017. The significant role of aromatics highlights the importance of setting up corresponding measures to suppress ozone formation.

We present the components driving the OFP growth by chemical group and sector from 1990-2017 in Fig. 5. As illustrated above, during 1990-2000, the rapidly increasing number of vehicles along with economic development introduced large quantities of reactive alkenes and aromatics, as well as OFP (+45%). From 2000 to 2010, solvent use became the largest
contributor to the OFP change, accompanied by the boom in aromatics emissions. During this period, on-road gasoline vehicles and industry also played important roles in the OFP increase caused by alkenes. The OFP contribution of alkanes was small even though the emission increase was significant, due to their low chemical reactivity. Notably, during the most recent years (2010-2017), emissions of several key source categories have started to stabilize or even decrease, significantly mitigating the increased OFP caused by solvent use and industrial processes. OFP reductions are mainly attributed to alkenes
(-3.3 Tg-$O_3$) and OVOC (-2.5 Tg-$O_3$) associated with the sectors of residential biofuel combustion and transportation.

By allocating the emissions into grids based on spatial surrogates, we depicted the spatial distributions of OFP at a spatial resolution of 0.25° × 0.25° in Fig. 6. Power plant locations derived from CPED and verified using Google Earth were used to distribute the corresponding emissions; rural population density and roadmaps were used as proxies for residential biofuel combustion and on-road vehicles, respectively; urban and total population datasets were used to allocate the emissions of
other sources. The spatial patterns of OFP are in general consistent with the observed ozone maps (see Fig. 6). Significant signals of urbanization are especially prevalent over China, demonstrating the potential severe ozone pollution in densely populated regions such as eastern China and the Sichuan basin. We further analyzed the OFP trends for four key regions with severe ozone pollution in China, i.e., Beijing-Tianjin-Hebei (BTH), Yangtze River Delta (YRD), Pearl River Delta



(PRD) and the Sichuan Basin (SCB) for 2013-2017. Compared to the values of 2013, OFP for BTH, YRD, PRD and SCB showed minor growth ratio of 2%, 8%, 9% and 2% in 2017, respectively. Especially, the mitigation measures covering various sources implemented in PRD have achieved a 7% decrease of OFP in 2014-2017. This suggests that due to the absence of effective mitigation measures regarding NMVOC, the potential ozone production for megacity clusters have been

stable in recent years, in contrast to the dramatic reductions of $NO_x$ and primary $PM_{2.5}$.

We depicted the trend of summer mean (June-July-August) maximum daily 8-hour average (MDA8) ozone concentrations observed by the ground monitoring network over China since 2013 in Fig. 6. Different from the flat trend of OFP during the same period, significant increases in ozone were observed in northern, central and southwest China. Previous studies indicated that VOC-limited conditions prevail in megacities of China, where more ozone can be produced as a result of

dramatic $NO_x$ emission decline (Gao et al., 2017; Li K et al., 2018). The effective reduction of $PM_{2.5}$ in recent years favoring the penetration of ultraviolet light to the surface may also lead to greater ozone production. Based on model simulation, Li K et al. (2018) demonstrates that the sharp decrease of $NO_x$ emissions (~21%) in VOC-limited regions, and the ~40% reduction of $PM_{2.5}$ slowing down the aerosol sink of hydro-peroxy, drove the rise of surface ozone in China during 2013-2017. As indicated by above analyses, the lagging behind mitigation measures of NMVOC compared to other criteria pollutants in

most urban China have advanced ozone production through non-linear chemistry in the gas phase and/or multiphase chemistry between gases and aerosols. Designing cost-effective mitigation measures for NMVOC accompanying with $NO_x$, CO and $PM_{2.5}$ is quite urgent and crucial for ozone control in the near future.

## 4    Discussion

### 4.1   Comparison with previous studies

The NMVOC emissions in China estimated in our work are compared with previous estimates in Fig. 7. The increasing pattern since 2000 is generally consistent among different long-term emission inventories (Kurokawa et al., 2013, Wu et al., 2016). Our estimates are slightly lower than the values of Wu et al. (2016), Wu and Xie (2017) and Regional Emission inventory in Asia version 2.1 (REAS v2.1, Kurokawa et al., 2013) but are higher than those of Wei et al. (2014) and Bo et al. (2008). The emissions estimated by the various inventories for the most recent years, i.e., since 2010, agree well, with

variations of <13%. For regional emission inventories, as the data sources of activity rates are generally consistently obtained from official statistics, we can attribute the emission differences to the distinct source classification system and assignment of emission factors. For the emission inventory at a global scale, the calculated growth rate of NMVOC emissions in Emissions Database for Global Atmospheric Research (EDGAR, EC-JRC/PBL, 2011; Crippa et al., 2018) is much slower than our estimates. EDGAR emissions are slightly lower than our estimates after 2000 but much higher in the

1990s. The reasons for these differences are complicated and should include inconsistency in source categories, data sources of activity rates and emission factors (Li M et al., 2018).



## 4.2 Uncertainty analyses

The uncertainties in the total NMVOC emissions in China were estimated to be at a moderate level of ±68% ~ ±78% (Zhang et al., 2009; Kurokawa et al., 2013), mainly arising from the lack of reliable data for scattered areal sources. On the other hand, selection and application of source profiles can lead to over three magnitudes of differences in quantifying individual species, representing the largest uncertainty sources during speciation (Li et al., 2014).

Uncertainties in the composite profile are related to the accuracy of individual profiles applied for speciation. We calculated the uncertainties in the composite ones via the propagation of errors approach. For each species included in sources, the standard error (SE) for all profiles that had measurements were calculated to represent the "true" mass fraction error based on limited samples. If only one profile is used, expert judgement was used to estimate the profile error. We assumed the coefficients of variation (CVs, i.e., standard deviation / mean) of the SPECIATE v4.5 profiles with overall qualities of A~E to be ±5%~500%, and assigned local source profiles at CVs of ±5% ~ ±15% according to the measurement year based on expert judgment. Errors were added linearly for sources that shared profiles and then combined in quadrature into subsectors. Then the profile uncertainties were calculated to be 1.96 times the CV at the 95% confidence interval.

Source profiles are still the largest sources of uncertainty in determining emissions of individual chemical species and further species-specific OFP (see Fig. S1). Despite using similar sampling and analytical methods, the measured profiles show significant diversity among different studies, varying with fuel type, combustion technology, end-of-pipe control facilities, solvent components, etc. For the abundant components of toluene, ethylene, m- and p-xylene and propylene, the uncertainties in the mass fractions are in the range of 7% ~ 453%, with averages of 74% ~ 101%, showing comparable accuracy to the emission estimates. Profiles of vehicular sources and paint use show low uncertainties for all chemical species, including on-road gasoline, on-road diesel, off-road diesel and industrial paint use, because of the inclusion of reliable local source profiles. The uncertainty matrix shown in Fig. S1 highlights the need for more measurements and further analyses for important sources (species), especially chemical industry (o-xylene, benzene), other industrial processes (toluene, xylenes, formaldehyde, 2-methyl-2-butene), and residential biofuel combustion (toluene, xylenes, ethylbenzene, cis-2-butene, butyl cellosolve). Yet, it is difficult to quantify the uncertainties for trace gases such as polycyclic aromatic hydrocarbons (PAHs) and halo-hydrocarbons that have high molar mass but have not been included in standard sampling and analytical protocols in current studies.

The inter-annual variations in profiles are yet to be investigated, and more local source profiles with complete information of chemical species are still in need. An improved linkage between the source profile matrix and the source, province and temporal information will be important to improve the accuracy of emission estimates.

## 4.3 Policy implications

Ozone pollution has become increasingly severe in China, especially in megacities (Gao et al., 2017; Lu et al., 2018). Both ground measurements and satellites have detected an increasing trend of tropospheric ozone concentrations over recent



decades due to emissions of precursors, tropospheric chemistry, penetration from the stratosphere and meteorological changes (Verstraeten et al., 2015; Wang et al., 2016; Li K et al., 2018).

Our estimates of the speciated NMVOC emissions and underlying indications will be important for establishing the most cost-effective mitigation measures for ozone. The effective actions to control fine PM in China have gained significant achievement since 2013, while the ozone problem has not been fully addressed by the Chinese government. Based on our estimates, the implementation of control measures for vehicles, industrial and residential sources have led to emission reductions of alkenes, alkanes and aldehydes. However, due to the absence of effective control for evaporative sources, large amounts of aromatics have been emitted from the condensed phase into the atmosphere since 2000. Paint use, chemical industry, petroleum production and distribution, and other solvent use sources are the main contributors to the changes since 2010 and remain inefficiently controlled nationwide. These sources should be addressed and controlled more stringently in the next step. As urban China are mainly in VOC-limited conditions, mitigation of NMVOC emissions will suppress ozone formation effectively. China urgently needs to set up goals and enact more stringent legislation to control NMVOC emissions along with $NO_x$, CO and $PM_{2.5}$ to prevent further potential ozone pollution and adverse effects on human health.

## 5    Concluding remarks

Long-term speciated NMVOC emissions over China were estimated based on the MEIC framework and an updated local source profile database for the period of 1990 and 2017. Our results showed that China's emission of NMVOC increased by 192% from 9.76 Tg in 1990 to 28.5 Tg in 2017, due to the economic development and relatively late implementation of the emission control strategy. From 1990 to 2017, industrial sources and solvent use were the main driving forces for the emission increment, while the reduction of residential biofuel use and on-road vehicle exhaust in recent years lowered the rapid growth rates. Consequently, toluene and xylene emissions increased by more than a factor of 6 and surpassed those of ethylene. The proportion of aromatics emissions increased monotonically from 20% in 1990 to 33% in 2017, becoming the largest contributor in China at present. Meanwhile, the emissions of alkanes (e.g., ethane), alkenes (ethylene, propylene) and OVOC (formaldehyde) showed decreasing trends during 2010-2017.

The persistent growth of NMVOC emissions has led to increasingly enhanced ozone production over the last two decades but tend to stabilize in recent years. The total OFP in China was estimated to have dramatically increased from 38.2 Tg-$O_3$ in 1990 to 99.7 Tg-$O_3$ in 2017, with distinct driving sources in different economic development periods. On-road gasoline vehicles, paint use and industrial sources were the major contributors to the dramatic increases of emission and OFP from 1990-2010. Large amounts of OFP produced by reactive aromatics and alkenes were estimated during this period. For 2010-2017, OFP increased by only 7%, attributed to the implementation of successful clean air policies covering the transportation and industry sectors, as well as the reduced biofuel use in residential stoves. In 2017, the national OFP value was dominated by aromatics (43%) and alkenes (37%). Controlling the emissions of aromatics and alkenes from solvent use and industrial processes is crucial to addressing the ozone problem.



Ozone pollution has become a severe problem over megacity clusters in China. Due to the absence of effective control measures regarding NMVOC, OFP has shown stable trends in China since 2013, in contrast to the dramatic emission reductions for $NO_x$, CO and primary $PM_{2.5}$. Discrepancies between the increase of observed surface $O_3$ and the flat OFP trends might be the results of nonlinear chemistry and multiphase chemistry caused by this imbalance. Controlling NMVOC

5    emissions is anticipated to be efficient to suppress ozone formation because VOC-limited conditions prevail most urban areas in China. Considering the potential adverse effects on human health and complicated production mechanisms for ozone in the troposphere, China urgently needs to formulate ozone control policies based on the updated source information for precursors, including $NO_x$, CO and NMVOC, and setting up cost-effective measures to mitigate both $PM_{2.5}$ and ozone.

10    ***Author contribution.*** QZ designed the research. ML, QZ, BZ, DT, YL, FL, CH, SK, LY, and YZ calculated total NMVOC emissions. ML developed speciated VOC emissions and estimated OFP. ML, QZ, YZ, HS, YC, YB and KH interpreted the data. ML and QZ wrote the manuscript with input from all co-authors.

***Acknowledgements.*** This work was supported by the National Key R&D program (2016YFC0201506) and the National

15    Natural Science Foundation of China (91744310, 41625020, 41571130035, and 41571130032).



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



**Figure 1. NMVOC emissions in China for the period of 1990 to 2017.**



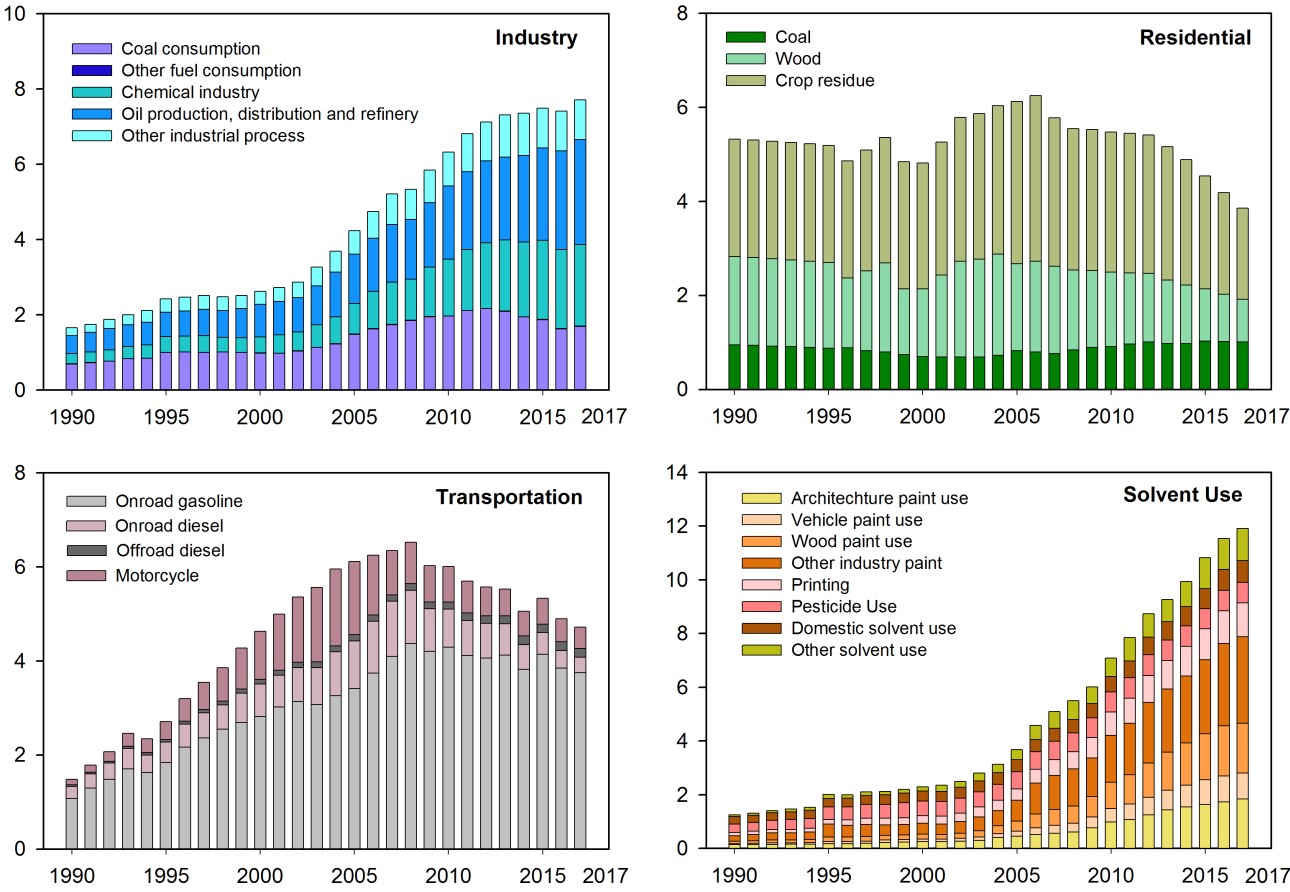

**Figure 2. NMVOC emissions by sub-categories of industry, residential, transportation and solvent use sector in China from 1990 to 2017 (Unit: Tg).**





**Figure 3. Emissions of the top 30 species contributing to OFP in 2017 in a OFP-descending order by sector in 1990 (a), 2000 (b), 2010 (c), and 2017 (d).**







**Figure 4. Emission trends for key chemical species (ethane, ethylene, toluene, xylene, formaldehyde, acetylene) during 1990-2017.**

**"Xylene" includes all isomers of xylene (m & p –xylene, o-xylene).**





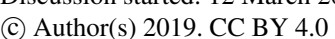

**Figure 5. Decomposed changes by chemical group and sector for emission and OFP from 1990 to 2017.**



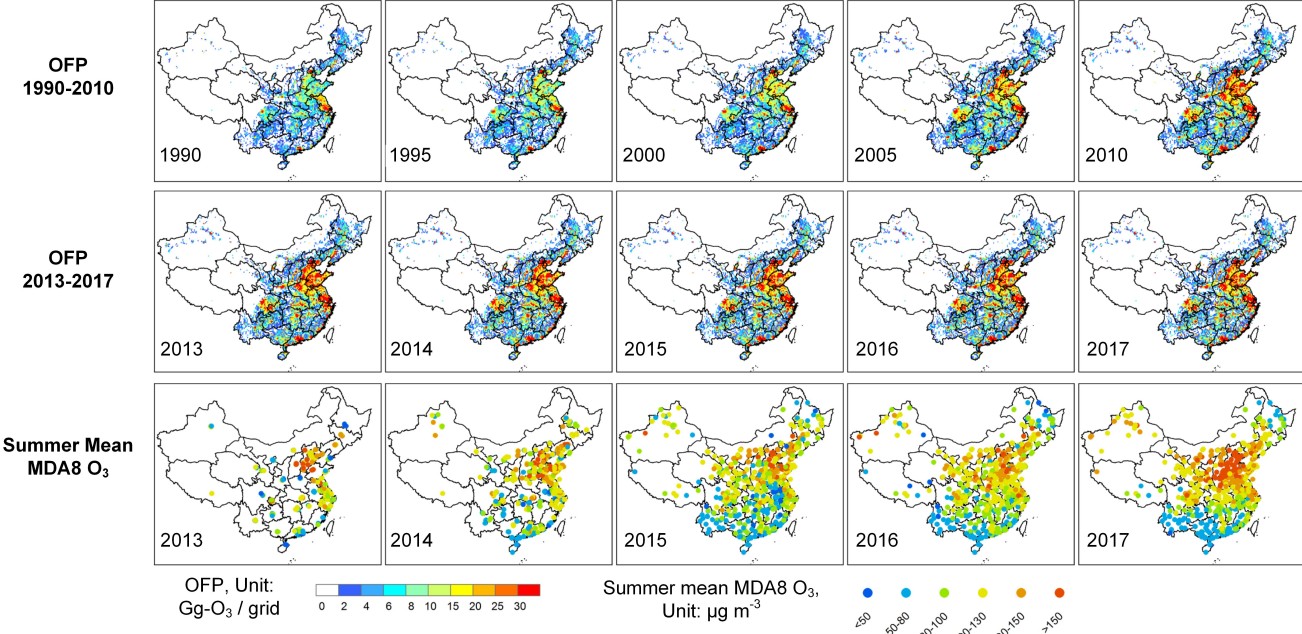

**Figure 6. OFP spatial distributions during 1990-2017 (the 1st and 2nd panel), and the observed summer mean (June-July-August) maximum daily 8-hour average (MDA8) surface ozone concentrations in 2013-2017 (the 3rd panel).**





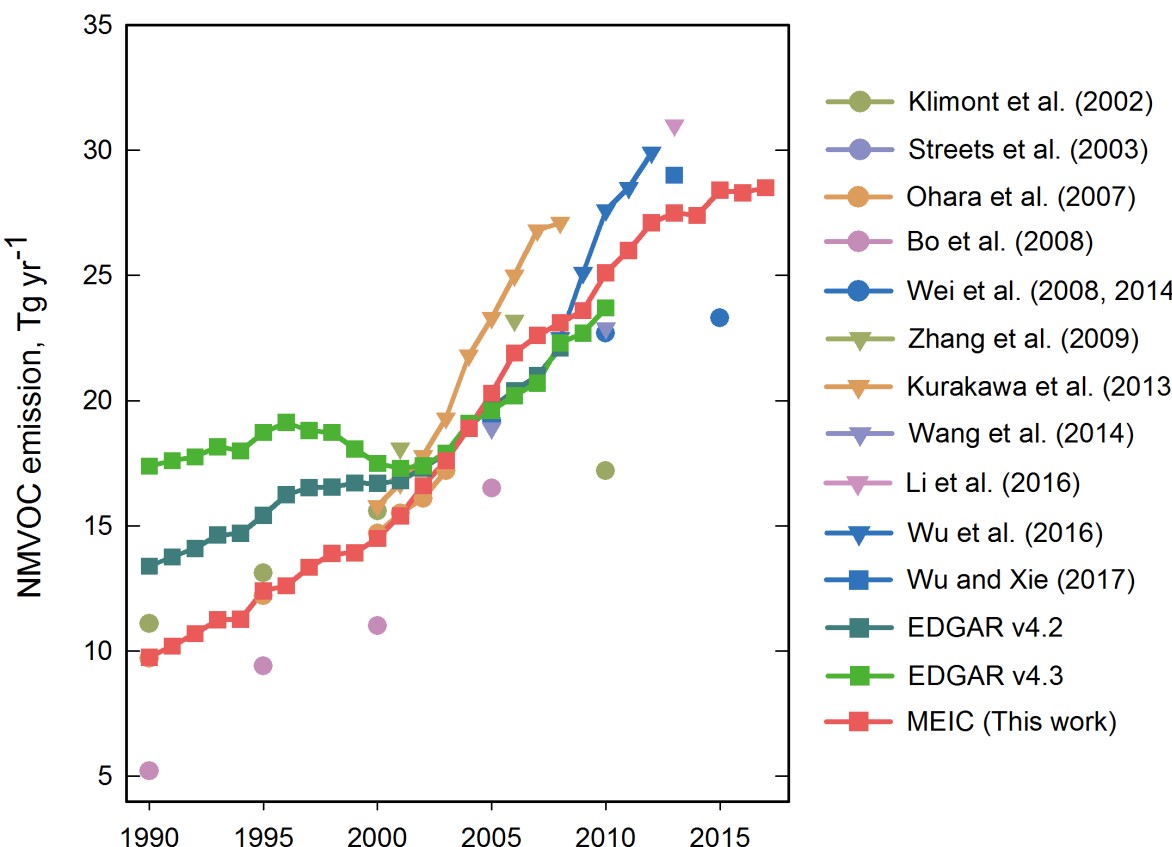

**Figure 7. Comparisons of NMVOC emissions in China between this work and previous studies.**



**Table 1. NMVOC emissions and OFP by source categories.**

| Sector | Sub-sector | NMVOC emission, Gg | | | | | | OFP, Gg-O$_3$ | | | | | |
|---|---|---|---|---|---|---|---|---|---|---|---|---|---|
| | | 1990 | 2000 | 2005 | 2010 | 2013 | 2017 | 1990 | 2000 | 2005 | 2010 | 2013 | 2017 |
| Power | **Power** | **8.0** | **14.3** | **28.0** | **42.3** | **50.2** | **54.0** | **25.5** | **44.5** | **86.0** | **126.4** | **149.6** | **159.7** |
| Industry | Chemical industry | 277.1 | 428.0 | 810.2 | 1513.2 | 1891.2 | 2177.0 | 234.2 | 514.3 | 942.4 | 1797.4 | 2314.8 | 3016.1 |
| | Industrial coal use | 688.4 | 977.1 | 1478.7 | 1958.3 | 2089.1 | 1684.7 | 2354.3 | 3292.6 | 5059.7 | 6873.5 | 7266.9 | 5739.9 |
| | Industrial other fuel combustion | 2.8 | 3.4 | 4.7 | 5.2 | 5.5 | 6.4 | 11.0 | 12.1 | 15.8 | 17.1 | 17.0 | 18.9 |
| | Oil production distribution and refinery | 469.7 | 871.1 | 1312.6 | 1937.5 | 2205.9 | 2788.8 | 1362.0 | 2562.9 | 3982.2 | 5958.3 | 6835.0 | 8845.2 |
| | Other industrial process | 210.1 | 338.1 | 626.6 | 903.3 | 1115.4 | 1050.0 | 309.2 | 519.5 | 1030.8 | 1544.6 | 1885.1 | 1801.0 |
| | **Sum of Industry** | **1648.1** | **2617.6** | **4232.8** | **6317.4** | **7307.1** | **7707.0** | **4270.6** | **6901.5** | **11031.0** | **16191.0** | **18318.9** | **19421.1** |
| Residential | Residential coal combustion | 949.9 | 697.1 | 826.3 | 914.0 | 976.5 | 1009.0 | 4219.5 | 3078.0 | 3629.6 | 3999.5 | 4283.9 | 4421.1 |
| | Residential biofuel combustion | 4370.7 | 4115.7 | 5298.1 | 4555.5 | 4183.2 | 2846.9 | 20543.7 | 19623.9 | 25270.2 | 21735.6 | 20040.6 | 13650.0 |
| | Residential other fuel combustion | 0.4 | 1.2 | 2.4 | 4.5 | 5.3 | 7.9 | 1.3 | 3.7 | 7.9 | 15.0 | 18.1 | 27.1 |
| | Waste treatment | 48.6 | 84.2 | 107.1 | 150.2 | 164.6 | 192.4 | 71.4 | 123.7 | 157.7 | 221.8 | 244.4 | 287.2 |
| | **Sum of Residential** | **5369.6** | **4898.2** | **6233.9** | **5624.2** | **5329.6** | **4056.2** | **24835.9** | **22829.3** | **29065.4** | **25972.0** | **24587.0** | **18385.4** |
| Solvent use | Industrial paint use | 466.1 | 940.5 | 1793.6 | 4202.5 | 5932.9 | 7879.1 | 1628.7 | 3354.3 | 6530.4 | 15877.9 | 22679.5 | 30159.4 |





| | | | | | | | | | | | | |
|---|---|---|---|---|---|---|---|---|---|---|---|---|
| | Solvent use other than paint | 788.2 | 1354.3 | 1886.7 | 2884.5 | 3338.6 | 4031.2 | 1284.1 | 2284.2 | 3244.1 | 5606.6 | 6631.7 | 7872.5 |
| | **Sum of Solvent use** | **1254.3** | **2294.8** | **3680.3** | **7086.9** | **9271.5** | **11910.3** | **2912.8** | **5638.5** | **9774.5** | **21484.5** | **29311.2** | **38031.9** |
| Transpor tation | On-road gasoline | 1188.2 | 3836.0 | 4962.9 | 5047.8 | 4683.7 | 4207.4 | 4868.3 | 16577.3 | 23206.0 | 25043.0 | 23668.9 | 21401.2 |
| | On-road diesel | 257.2 | 691.6 | 1022.6 | 809.5 | 665.0 | 330.5 | 1118.6 | 3008.7 | 4294.5 | 3350.7 | 2751.5 | 1374.5 |
| | Off-road diesel | 31.3 | 99.3 | 130.2 | 150.7 | 175.1 | 184.7 | 171.3 | 514.4 | 681.2 | 799.4 | 912.0 | 952.9 |
| | **Sum of Transportation** | **1476.7** | **4626.9** | **6115.7** | **6008.0** | **5523.8** | **4722.5** | **6158.3** | **20100.5** | **28181.7** | **29193.1** | **27332.4** | **23728.5** |
| **Sum of all sectors, Tg** | | **9.8** | **14.5** | **20.3** | **25.1** | **27.5** | **28.5** | **38.2** | **55.5** | **78.1** | **93.0** | **99.7** | **99.7** |