# Peer review of "Persistent growth of anthropogenic NMVOC emissions in China during 1990-2017: dynamics, speciation, and ozone formation potentials"

_Atmospheric Chemistry and Physics, 2019_

## Referee Comment (RC1) · Anonymous Referee #1 · 10 Apr 2019

This manuscript presents a comprehensive bottom-up inventory of China's NMVOC compounds between 1990 and 2017, including detailed information on sectors and speciation. The causes of trends in total NMVOC emissions and in specific sectors, including economic factors and pollution control strategies, are discussed. The impacts of NMVOC emissions changes on ozone formation potentials in China are quantified. The policy implications of China's NMVOC emissions changes are highlighted.

The manuscript is well presented and informative. The datasets presented here will be very useful to the atmospheric research and environmental policymaking communities. The manuscript is a valuable contribution to the literature. I recommend publication

after some issues are addressed.

Detailed Comments:

Title: - Suggest changing "dynamics" to "trends". "Dynamics" has a specific meaning in atmospheric science that is unrelated to emissions. I think the authors simply mean the trends in emissions.

Methods: - Emissions from open biomass burning were excluded from the inventory. Does this sector include crop burning? How big of a potential source is open biomass burning across China? What are the implications of excluding it from the inventory - Are NMVOC emissions from shipping in ports and near coastal areas included in the inventory?

Results: - Solvent use (including both industrial and residential sources) is now the largest NMVOC sector in China. How certain are the estimated emissions in this sector? McDonald et al. [Science, 2018] recently showed that, for cities in the United States, there are large differences between currently bottom-up approaches for estimating NMVOC emissions from paints, adhesives, and other sources lumped in the "solvent use" category of this manuscript. Should we expect similar uncertainties in Chinese solvent use emissions? If so, what are the implications for uncertainties in Chinese NMVOC speciation from solvent use, and for the resulting ozone formation potentials?

Discussion: - P 11, L 24-25: The differences between inventories in the most recent years appearsto be greater than 13% (for example comparing the 2014 values from MEIC and Wei inventories). Please change this sentence to more accurately reflect the data presented in Figure 8. - P 12, L 24: There are also large uncertainties for emissions of many compounds emitted from the Waste Treatment sector. Please note this fact.

References: - Liu et al., 2015, is missing from the list of references. Please include it.

Data accessibility: - For this paper to be useful to the community, the detailed inventory datasets reported here must be publicly available. A clear statement is needed in the manuscript about how the community can obtain the annual national and gridded emissions datasets and the sectoral and speciated detailed data presented here.

Technical Comments:

The figures and tables are clear and easy to follow. I have no changes to suggest for these.

The manuscript is also generally well written. However, there are numerous small mistakes in English grammar and usage throughout. For example, on the first page alone, I found the following errors: - P 1, L 20: omit "," after "that" - P 1, L 20: omit "been" - P 1, L 22: omit "," after "that" - P 1, L 24: change "offset" to "offsetting" - P 1, L 27: change "form" to "from" - P 1, L 28: change "were" to "was" - P 1, L 31: change "increase" to "increasing" There are similar small errors throughout the manuscript. Please correct them before resubmitting the revised manuscript.

---

## Referee Comment (RC2) · Anonymous Referee #2 · 10 May 2019

VOCs are key precursors of SOA and O3, and their emissions are of great uncertainty compared to some other species like SO2 or NOX, attributed to complicated sources and relative lack of field measurements. This paper presents a comprehensive analysis on China's national VOC emissions from 1990 to 2017, by source category and chemical component. It provides a very clear picture of the inter-annual trend, speciation variation, and the driving force of VOC emissions for the country. I only have some small concerns on the explanation of specific data and results and detailed comments follow. I suggest its publication subjected to minor revisions.

1. Table S1 in the supplement summarized the emission factors and activity levels by

source category. What are the meanings of the numbers in Column E (source profile)? In Column J, it seems that most of emission factors still came from foreign studies? Does that mean recent progress on local emission factors was very limited? I suggest the authors make some discussions here.

2. It is very interesting to know the control strategy and benefits of VOC emissions, as limited information was reported in previous inventories. I expected the VOC control started later than SO2 or NOX control. Relevant information is given in the last part of Section 2.1. Here I suggest the authors highlight the information in, for example, Table S1, thus the audience could understand the control strategy more clearly. Current table include only unabated emission factors.

3. It seems that open biomass burning is not included in the emission estimation. Could it be a potential bias of the estimate? Some review and discussion should be given.

4. More description in Figure 5 should be provided in the caption. What are the meanings of the species indicated for each year? The species contributing most to the variation of emissions/OFP?

---

## Author Comment (AC1) · 18 Jun 2019

**Response to referee #1**

*This manuscript presents a comprehensive bottom-up inventory of China's NMVOC compounds between 1990 and 2017, including detailed information on sectors and speciation. The causes of trends in total NMVOC emissions and in specific sectors, including economic factors and pollution control strategies, are discussed. The impacts of NMVOC emissions changes on ozone formation potentials in China are quantified. The policy implications of China's NMVOC emissions changes are highlighted.*

*The manuscript is well presented and informative. The datasets presented here will be very useful to the atmospheric research and environmental policymaking communities. The manuscript is a valuable contribution to the literature. I recommend publication after some issues are addressed.*

**Response:** We thank the positive and constructive comments given by the referee #1, which are very helpful to improve the manuscript. Our response to each specific comment is presented below.

Detailed Comments and Responses:

*1. Title: - Suggest changing "dynamics" to "trends". "Dynamics" has a specific meaning in atmospheric science that is unrelated to emissions. I think the authors simply mean the trends in emissions.*

**Response:** Thanks for the suggestion. We change the "dynamics" to "drivers" in the title, to represent the analyses on emission trends and the underlying driving forces.

*2. Methods: - Emissions from open biomass burning were excluded from the inventory. Does this sector include crop burning? How big of a potential source is open biomass burning across China? What are the implications of excluding it from the inventory - Are NMVOC emissions from shipping in ports and near coastal areas included in the inventory?*

**Response:** We exclude open burning of biomass (including crop open burning in the field), and include the combustion of household biofuel (crop residue, wood) in our inventory. Emissions of open biomass burning are always estimated separately from the anthropogenic sector because distinct method and dataset is applied (Wiedinmyer et al., 2011; Huang et al., 2012; Randerson et al., 2018; Yin et al., 2019). Based on the most recent work (Yin et al., 2019), emissions of open biomass burning in China are 1.12~2.16 Tg NMHC, corresponding to 2.90~5.60 Tg NMVOC by applying an averaged OVOC fraction of 61.4% (8753 in SPECIATE 4.5, Andreae and Merlet, 2001) during 2003-2017. Compared to 17.6~28.5 Tg NMVOC from anthropogenic sources during the same period, we capture > 83% of the total emissions when including open biomass burning in the analyses, and the total NMVOC emission will be 32.8 Tg in 2017, with emission decrease for 2015-2016. We add more analyses in the discussion section of the revised manuscript.

NMVOC emissions from international shipping are also excluded in the inventory. We clarify

the sources included in the inventory in the revised manuscript.

*3. Results: - Solvent use (including both industrial and residential sources) is now the largest NMVOC sector in China. How certain are the estimated emissions in this sector? McDonald et al. [Science, 2018] recently showed that, for cities in the United States, there are large differences between currently bottom-up approaches for estimating NMVOC emissions from paints, adhesives, and other sources lumped in the "solvent use" category of this manuscript. Should we expect similar uncertainties in Chinese solvent use emissions? If so, what are the implications for uncertainties in Chinese NMVOC speciation from solvent use, and for the resulting ozone formation potentials?*

**Response:** Thanks for the referee's comments. Uncertainties of emission estimates are always difficult to quantify because of the lack of statistics and measurements. For solvent use, high uncertainties are expected, considering the numerous scattered areal sources included in this sector, uncertainties in statistics, complex technologies and emission factors. According to the uncertainty assessment of Wu et al. (2016) using the Monte Carlo simulation, the uncertainty of the solvent use sector is high up to -70% ~ 202% in 2012. This uncertainty in the total NMVOC emissions will propagate to the speciation results and also the ozone formation potentials. For solvent use, the uncertainty in source profiles are estimated as 110% (average, 9.8% ~ 973%, for top 30 chemical species, as shown in Fig. S1). Thus a high uncertainty of 130%~230% can be roughly estimated for the results of speciation and OFPs for the solvent use sector.

*4. Discussion: - P 11, L 24-25: The differences between inventories in the most recent years appears to be greater than 13% (for example comparing the 2014 values from MEIC and Wei inventories). Please change this sentence to more accurately reflect the data presented in Figure 8. - P 12, L 24: There are also large uncertainties for emissions of many compounds emitted from the Waste Treatment sector. Please note this fact.*

**Response:** Thanks for the comments. We revise the sentences as below:

"The emissions estimated by the various inventories for the most recent years, i.e., since 2010, agree relatively well, with variations of 10%~22%."

"The uncertainty matrix shown in Fig. S1 highlights the need for more measurements and further analyses for important sources (species), especially chemical industry (o-xylene, benzene), other industrial processes (toluene, xylenes, formaldehyde, 2-methyl-2-butene), residential biofuel combustion (toluene, xylenes, ethylbenzene, cis-2-butene, butyl cellosolve), and waste treatment (xylenes, ethylene, formaldehyde)."

*5. References: - Liu et al., 2015, is missing from the list of references. Please include it.*

**Response:** Added.

*6. Data accessibility: - For this paper to be useful to the community, the detailed inventory datasets reported here must be publicly available. A clear statement is needed in the manuscript*

*about how the community can obtain the annual national and gridded emissions datasets and the sectoral and speciated detailed data presented here.*

**Response:** We thank the referee for the suggestion. We have uploaded the data presented in the manuscript to a public repository and added a statement in the "Data availability" section.

Technical Comments and Responses:

*7. The figures and tables are clear and easy to follow. I have no changes to suggest for these.*

*The manuscript is also generally well written. However, there are numerous small mistakes in English grammar and usage throughout. For example, on the first page alone, I found the following errors: - P 1, L 20: omit "," after "that" - P 1, L 20: omit "been" - P 1, L 22: omit "," after "that" - P 1, L 24: change "offset" to "offsetting" - P 1, L 27: change "form" to "from" - P 1, L 28: change "were" to "was" - P 1, L 31: change "increase" to "increasing" There are similar small errors throughout the manuscript. Please correct them before resubmitting the revised manuscript.*

**Response:** We thank the referee's careful reading and the detailed comments. We carefully correct the mistakes throughout the manuscript.

**References**

Andreae, M. O., and Merlet, P.: Emission of trace gases and aerosols from biomass burning, Global Biogeochemical Cycles, 15, 955-966, 10.1029/2000GB001382, 2001.

Huang, X., Li, M., Li, J., and Song, Y.: A high-resolution emission inventory of crop burning in fields in China based on MODIS Thermal Anomalies/Fire products, Atmospheric Environment, 50, 9-15, https://doi.org/10.1016/j.atmosenv.2012.01.017, 2012.

Randerson, J.T., G.R. van der Werf, L. Giglio, G.J. Collatz, and P.S. Kasibhatla. 2018. Global Fire Emissions Database, Version 4, (GFEDv4). ORNL DAAC, Oak Ridge, Tennessee, USA. https://doi.org/10.3334/ORNLDAAC/1293

Wiedinmyer, C., Akagi, S. K., Yokelson, R. J., Emmons, L. K., Al-Saadi, J. A., Orlando, J. J., and Soja, A. J.: The Fire INventory from NCAR (FINN): a high resolution global model to estimate the emissions from open burning, Geosci. Model Dev., 4, 625-641, 10.5194/gmd-4-625-2011, 2011.

Wu, R., Bo, Y., Li, J., Li, L., Li, Y., and Xie, S.: Method to establish the emission inventory of anthropogenic volatile organic compounds in China and its application in the period 2008–2012, Atmospheric Environment, 127, 244-254, http://dx.doi.org/10.1016/j.atmosenv.2015.12.015, 2016.

Yin, L., Du, P., Zhang, M., Liu, M., Xu, T., and Song, Y.: Estimation of emissions from biomass burning in China (2003–2017) based on MODIS fire radiative energy data, Biogeosciences, 16, 1629-1640, 10.5194/bg-16-1629-2019, 2019.

---

## Author Comment (AC2) · 18 Jun 2019

**Response to referee #2**

*VOCs are key precursors of SOA and O3, and their emissions are of great uncertainty compared to some other species like SO2 or NOX, attributed to complicated sources and relative lack of field measurements. This paper presents a comprehensive analysis on China's national VOC emissions from 1990 to 2017, by source category and chemical component. It provides a very clear picture of the inter-annual trend, speciation variation, and the driving force of VOC emissions for the country. I only have some small concerns on the explanation of specific data and results and detailed comments follow. I suggest its publication subjected to minor revisions.*

**Response:** We thank the positive comments given by referee #2, which are very helpful to improve the manuscript. Our response to each specific comment is presented below.

Detailed Comments and Responses:

*1. Table S1 in the supplement summarized the emission factors and activity levels by source category. What are the meanings of the numbers in Column E (source profile)? In Column J, it seems that most of emission factors still came from foreign studies? Does that mean recent progress on local emission factors was very limited? I suggest the authors make some discussions here.*

**Response:** Column E of Table S1 represent the source profiles used for each source category during speciation. The numbers are the "P_NUMBER" of profiles in the SPECIATE v4.5 database. We add a note in Table S1 to make it clearer.

As illustrated in the main text, we firstly evaluated the emission factors based on local measurements or determined by taking China's regulations into account, e.g., the values of Wei et al. (2009) for solvent use, Tsai et al. (2003) for residential coal combustion, and the technology-based emission factors derived from Zheng et al. (2014) for on-road vehicles. For sources that lack reliable local emission factors, we mainly refer to European studies (EEA, 2016) or AP-42 (EPA, 1995), combined with source information from local investigations where available (Zhang et al., 2000; Tsai et al., 2003; He, 2006; Li et al., 2011; Wang et al., 2013).

In recent years, more and more local emission factors are measured, covering biofuel combustion in stoves (Wang et al., 2009; Tsai et al., 2003; Zhang et al., 2000; Li et al., 2011), coal combustion in boiler and stoves (Tsai et al., 2003; Zhang et al., 2000), paint use (Wei et al., 2009), coke production (He, 2006), which have been used in compiling our inventory. For most industrial processes and solvent use sources, local measurements of emission factors are still limited and more investigations need to be conducted in the future. We add more discussions in the revised manuscript.

*2. It is very interesting to know the control strategy and benefits of VOC emissions, as limited information was reported in previous inventories. I expected the VOC control started later than SO2 or NOX control. Relevant information is given in the last part of Section 2.1. Here I suggest the authors highlight the information in, for example, Table S1, thus the audience could*

*understand the control strategy more clearly. Current table include only unabated emission factors.*

**Response:** Thanks for the comments. We highlight the control measures implemented for NMVOC emission control in Table S1.

*3. It seems that open biomass burning is not included in the emission estimation. Could it be a potential bias of the estimate? Some review and discussion should be given.*

**Response:** We add more discussion in the discussion section as follows:

"It should be noted that open biomass burning is excluded in the inventory, which may introduce bias for the total emission analyses covering all sources. Based on the most recent work (Yin et al., 2019), emissions of open biomass burning in China are 1.12~2.16 Tg NMHC, corresponding to 2.90~5.60 Tg NMVOC by applying an averaged OVOC fraction of 61.4% (8753 in SPECIATE 4.5, Andreae and Merlet, 2001) during 2003-2017. Compared to 17.6~28.5 Tg NMVOC from anthropogenic sources during the same period, we capture > 83% of the total emissions when including open biomass burning in the analyses, and the total NMVOC emission will be 32.8 Tg in 2017, with large emission decrease (-7%) for 2015-2016."

*4. More description in Figure 5 should be provided in the caption. What are the meanings of the species indicated for each year? The species contributing most to the variation of emissions/OFP?*

**Response:** The bars of species represent the contribution to the total emission / OFP changes for the specific time period. We revise the caption to make it clearer.

**References**

Andreae, M. O., and Merlet, P.: Emission of trace gases and aerosols from biomass burning, Global Biogeochemical Cycles, 15, 955-966, 10.1029/2000GB001382, 2001.

He, Q.: Characteristics, emission factors and emission estimation for particulate matters and volatile organic compounds emitted from coke production in China (in Chinese), Ph.D thesis, Guangzhou Institute of Geochemistry, Chinese Academy of Sciences, Guangzhou, 2006.

Li, X. H., Wang, S. X., and Hao, J.: Characteristics of Volatile Organic Compounds (VOCs) Emitted from Biofuel Combustion in China (in Chinese), Environ. Sci., 32, 3515-3521, 2011.

Tsai, S. M., Zhang, J., Smith, K. R., Ma, Y., Rasmussen, R. A., and Khalil, M. A. K.: Characterization of Non-methane Hydrocarbons Emitted from Various Cookstoves Used in China, Environ. Sci. Technol., 37, 2869-2877, doi: 10.1021/es026232a, 2003.

Wang, Q., Geng, C., Lu, S., Chen, W., and Shao, M.: Emission factors of gaseous carbonaceous species from residential combustion of coal and crop residue briquettes, Front. Environ. Sci. Eng., 7, 66-76, doi: 10.1007/s11783-012-0428-5, 2013.

Wang, S., Wei, W., Du, L., Li, G., and Hao, J.: Characteristics of gaseous pollutants from biofuel-stoves in rural China, Atmospheric Environment, 43, 4148-4154, http://dx.doi.org/10.1016/j.atmosenv.2009.05.040, 2009.

Wei, W., Wang, S., and Hao, J.: Estimation and forcast of volatile organic compounds emitted from paint uses in China (in Chinese), Environ. Sci., 30, 2809-2815, 2009.

Yin, L., Du, P., Zhang, M., Liu, M., Xu, T., and Song, Y.: Estimation of emissions from biomass burning in China (2003–2017) based on MODIS fire radiative energy data, Biogeosciences, 16, 1629-1640, 10.5194/bg-16-1629-2019, 2019.

Zhang, J., Smith, K. R., Ma, Y., Ye, S., Jiang, F., Qi, W., Liu, P., Khalil, M. A. K., Rasmussen, R. A., and Thorneloe, S. A.: Greenhouse gases and other airborne pollutants from household stoves in China: a database for emission factors, Atmo. Environ., 34, 4537-4549, doi: https://doi.org/10.1016/S1352-2310(99)00450-1, 2000.

Zheng, B., Huo, H., Zhang, Q., Yao, Z. L., Wang, X. T., Yang, X. F., Liu, H., and He, K. B.: High-resolution mapping of vehicle emissions in China in 2008, Atmos. Chem. Phys., 14, 9787-9805, doi: 10.5194/acp-14-9787-2014, 2014.